# Prevalence and determinants of anemia among Iranian population aged ≥35 years: A PERSIAN cohort–based cross-sectional study

Mohammad Zamani[1‡], Hossein Poustchi[2‡], Amaneh Shayanrad[2], Farhad Pourfarzi[3], Mojtaba Farjam[4], Kourosh Noemani[5], Ebrahim Ghaderi[6], Vahid Mohammadkarimi[7], Mahmood Kahnooji[8], Fariborz Mansour-Ghanaei[9], Ayoob Rastegar[10], Ali Mousavizadeh[11], Shideh Rafati[12], Masoumeh Ghoddusi Johari[13], Mahmood Moosazadeh[14], Alizamen Salehifardjouneghani[15,16], Alireza Ostadrahimi[17], Iraj Mohebbi[18], Alireza Khorram[19], Fatemeh Ezzodini Ardakani[20], Maryam Sharafkhah[2], Yahya Pasdar[21], Anahita Sadeghi[22], Reza Malekzadeh[23]*

1 Digestive Diseases Research Center, Digestive Diseases Research Institute, Tehran University of Medical Sciences, Tehran, Iran, 2 Liver and Pancreatobiliary Diseases Research Center, Digestive Diseases Research Institute, Tehran University of Medical Sciences, Tehran, Iran, 3 Digestive Disease Research Center, Ardabil University of Medical Sciences, Ardabil, Iran, 4 Noncommunicable Diseases Research Center, Fasa University of Medical Sciences, Fasa, Iran, 5 Department of Disease Prevention and Control, Deputy of Health Center, Ahvaz Jundishapur University of Medical Sciences, Ahvaz, Iran, 6 Social Determinants of Health Research Center, Research Institute for Health Development, Kurdistan University of Medical Sciences, Sanandaj, Iran, 7 Department of Internal Medicine, School of Medicine, Shiraz University of Medical Sciences, Shiraz, Iran, 8 Department of Internal Medicine, Faculty of Medicine, Rafsanjan University of Medical Sciences, Rafsanjan, Iran, 9 Gastrointestinal and Liver Diseases Research Center, Guilan University of Medical Sciences, Rasht, Iran, 10 Non-Communicable Diseases Research Center, Sabzevar University of Medical Sciences, Sabzevar, Iran, 11 Social Determinants of Health Research Center, Yasuj University of Medical Sciences, Yasuj, Iran, 12 Social Determinants in Health Promotion Research Center, Faculty of Health, Hormozgan University of Medical Sciences, Bandar Abbas, Iran, 13 Breast Diseases Research Center, Shiraz University of Medical Sciences, Shiraz, Iran, 14 Gastrointestinal Cancer Research Center, Non-Communicable Diseases Institute, Mazandaran University of Medical Sciences, Sari, Iran, 15 Modeling in Health Research Center, Shahrekord University of Medical Sciences, Shahrekord, Iran, 16 Department of Pediatrics, College of Medicine, Shahrekord University of Medical Sciences, Shahrekord, Iran, 17 Nutrition Research Center, Tabriz University of Medical Sciences, Tabriz, Iran, 18 Social Determinants of Health Research Center, Urmia University of Medical Sciences, Urmia, Iran, 19 Health Promotion Research Center, Zahedan University of Medical Sciences, Zahedan, Iran, 20 Department of Oral and Maxillofacial Radiology, Faculty of Dentistry, Shahid Sadoughi University of Medical Sciences, Yazd, Iran, 21 Research Center for Environmental Determinants of Health (RCEDH), Health Institute, Kermanshah University of Medical Sciences, Kermanshah, Iran, 22 Digestive Diseases Research Institute, Tehran University of Medical Sciences, Tehran, Iran, 23 Digestive Oncology Research Center, Digestive Diseases Research Institute, Tehran University of Medical Sciences, Tehran, Iran

‡ MZ and HP are contributed equally to this work as co-first authors.
* dr.reza.malekzadeh@gmail.com

## Abstract

### Background

So far, no comprehensive studies have been performed to assess burden and determinants of anemia in Iran. In the present study, we aimed to answer this query using the data obtained from the Prospective Epidemiological Research Studies in IrAN (PERSIAN).



**Data Availability Statement:** The study protocol and individual participant data that underlie the results reported in this study, after de-identification

(text, tables, and figures) can be shared with investigators whose proposed use of the data has been approved by the independent review committee of Tehran University of Medical Sciences and Digestive Diseases Research Institute. Data can be provided for projects related to the topic of the present study. The proposals should be directed to the PERSIAN cohort center (email: info@persiancohort.com), and/or Digestive Diseases Research Institute (email: info@ddri.ir), and/or Prof Reza Malekzadeh (email: dr.reza.malekzadeh@gmail.com), the senior author of the manuscript and the project leader.

**Funding:** The authors received no specific funding for this work.

**Competing interests:** The authors have declared that no competing interests exist.

## Methods

In this cross-sectional study, we included 161,686 adult participants (aged 35 years and older) from 16 provinces of Iran. Anemia was defined as a hemoglobin concentration of <13 g/dL in males and <12 g/dL in females. To evaluate the association between anemia and different factors, we used the multivariable Poisson regression analysis with robust variance by applying adjusted prevalence ratio (PR) with 95% confidence interval (CI).

## Results

Of the total number of subjects, 72,387 (44.77%) were male and others were female. Mean age was 49.39±9.15 years old. The overall age- and sex-standardized prevalence of anemia was 8.83% (95% CI: 8.70–8.96%) in the included population. The highest and the lowest age- and sex-standardized prevalence of anemia pertained to Hormozgan (37.41%, 95% CI: 35.97–38.85%) and Kurdistan (4.57%, 95% CI: 3.87–5.27%) provinces, respectively. Being female (PR = 2.97), rural residence (PR = 1.24), being retired (PR = 1.53) and housewife (PR = 1.11), third and fourth wealth status quartiles (PR = 1.09 and PR = 1.11, respectively), being underweight (PR = 1.49), drug user (PR = 1.35), inadequate sleep (PR = 1.16), poor physical activity (PR = 1.15), diabetes (PR = 1.09), renal failure (PR = 2.24), and cancer (PR = 1.35) were associated with increased risk of anemia. On the other hand, illiteracy (PR = 0.79) and abdominal obesity (PR = 0.77) decreased the risk of anemia.

## Conclusions

According to the results, a variable prevalence of anemia was observed across the included provinces. We tried to provide an informative report on anemia prevalence for health professionals and authorities to take measures for identification and management of the cases of anemia in high-prevalence areas.

## Introduction

Anemia is a multifactorial condition that is defined as an abnormal low red blood cell count or hemoglobin level, and it can be associated with a variety of serious health problems, such as severe fatigue and weakness, neurological impairment, cardiovascular diseases, etc. [1]. According to the Global Burden of Diseases, Injuries and Risk Factors report in 2013 (GBD 2013), about 1.93 billion people were affected by anemia around the world [2]. From 1990 to 2010, while anemia prevalence decreased slightly, the total number of cases increased. The burden was highest in under-5 years children and women. The years lived with disability (YLD) from anemia increased from 65.5 million years in 1990 to 68.4 million years in 2010 [3].

The most important cause of anemia is iron deficiency due to poor nutrition, menstruation or pregnancy [4]. For this reason, the prevalence of anemia is greater in developing countries [3, 5], and will be consequently associated with more healthcare demand and expenditure. Therefore, it would be useful to have an acceptable monitoring of the epidemiology of anemia in a developing country for better control of this disease. Iran, as a developing country, was reported by the World Health Organization (WHO) to have a moderate anemia prevalence [6]. Previous surveys from Iran reported variable prevalence rates of anemia in general

population (between 10% and 30%) [7]. However, no comprehensive studies have been reported for Iran yet.

In the present study, we aimed to use the data obtained from the Prospective Epidemiological Research Studies in IrAN (PERSIAN) [8] to assess the prevalence rate of anemia in a defined Iranian population. In addition, we tried to identify the risk factors for anemia, and the geographical differences in the prevalence of anemia in 16 provinces of Iran. Our results should be helpful for an appropriate future public health plan against anemia in Iran.

## Materials and methods

### Locations and patients

In this cross-sectional study, we performed analysis on the baseline information from the PERSIAN cohort study, including participants aged 35 years and older from 18 cohort centers in 16 provinces of Iran. The detailed information of the PERSIAN cohort has been explained elsewhere [8]. Briefly, this cohort study was launched in 2014 with the purpose of identifying the most common non-communicable diseases and the relevant risk factors among the adult Iranian population. For the present study, we used the baseline data of PERSIAN cohort aiming to determine the prevalence of anemia and the associated factors. The subjects were recruited in this study with a census sampling method. Those individuals with incomplete data of the necessary hematologic laboratory parameters were excluded from the study. The study protocol was approved by the ethics committee of Tehran University of Medical Sciences (IR. TUMS.DDRI.REC.1396.1). Written informed consent to participate in the survey was obtained from all participants.

### Data collection and measurements

To assess the determinants of anemia in the present study, we searched the literature and selected a number of factors potentially associated with the risk of the disease [9–11]. The required data were gathered from the participants by trained staff members using questionnaires consisting of demographic information and risk factors for anemia mentioned in the following:

- Demographic information, including sex (male, female), age, city, and residence (rural, urban).

- Socioeconomic variables, including educational level (illiterate, primary, secondary, tertiary), occupational status (unemployed, working, retired, housewife), and wealth score index.

- Individual factors, including body mass index (BMI), abdominal obesity (no, yes), ever cigarette smoker (no, yes), ever hookah smoker (no, yes), ever drug user (no, yes), ever alcohol user (no, yes), sleep duration (hours/day), and physical activity (good, poor).

- Past medical history, including diabetes (no, yes), hypertension (no, yes), renal failure (no, yes), cancer (no, yes), rheumatoid arthritis (no, yes), and lupus (no, yes).

Regarding the socioeconomic variables, the educational level was classified according to the years of education into four groups: illiterate (0), primary (1–6 years), secondary (7–12 years) and tertiary ($\geq$13 years). Wealth score index was calculated by the principal component analysis and categorized into four quartiles from poorest (1st quartile) to richest (4th quartile) [12].

With respect to the individual factors, the participants were grouped as underweight, normal weight, overweight and obese if their BMI was in the range of <18.5 kg/m$^2$, 18.5–24.9 kg/

m$^2$, 25–29.9 kg/m$^2$ and ≥30 kg/m$^2$, respectively. They were also known to have an abdominal obesity based on a high waist-to-hip ratio (>0.90 in males and >0.85 in females) [13]. Sleep duration was categorized as enough (7–9 hours/day), inadequate (<7 hours/day) and excessive (>9 hours/day) [14]. The physical activity was assessed using the Metabolic Equivalent Rates (METs), which is a self-report instrument for measuring the activities of daily living [15]. One MET is about 3.5 ml of oxygen consumed per kg per minute while sitting at rest. On a weekly basis, the mean MET rate of the subjects was calculated, which was 41 METs/hour/day. So, those individuals with less than 41 METs/hour/day were considered to have a poor physical activity.

Concerning the past medical history, diabetes was defined as a fasting plasma glucose ≥126 mg/dL, or consumption of glucose lowering medications, or a self-report of history of a physician-related diagnosis of diabetes. Hypertension was defined as a systolic blood pressure >140 mmHg or diastolic blood pressure >90 mmHg, or taking blood pressure lowering drugs, or a self-reported physician-related diagnosis of hypertension. Renal failure was defined by an estimated glomerular filtration rate of less than 60 mL/min/1.73 m$^2$, or being on dialysis, or reporting a past medical history of kidney transplantation [16].

In the present study, the anemia was defined as per the WHO criteria, that is, a hemoglobin concentration of <13 g/dL in males and <12 g/dL in females [6]. The anemia severity was categorized into three groups of mild (hemoglobin 11–12.9 g/dL in males and 11–11.9 g/dL in females), moderate (hemoglobin 8–10.9 g/dL), and severe anemia (hemoglobin <8 g/dL). Also, the anemia type was classified by the hematological indices of mean corpuscular hemoglobin concentration (MCHC) and mean corpuscular volume (MCV). The normal range for MCHC was considered as 32–36% and for MCV was considered as 80–100 fL [17].

## Statistical analysis

The database was first structured using Microsoft Office Excel software (Microsoft Corporation, Redmond, Washington). In addition, the analyses were performed using STATA software version 14 (StataCorp, College Station, TX, USA). Descriptive analysis and chi-squared test were used to calculate frequency and percentage of anemia distribution by categorical variables. Both of crude and age- and sex-standardized prevalence rates were reported for anemia. The age- and sex-standardized prevalence rates were calculated using direct method, and Iran national census in 2016 were considered as a standard population. We used a complex survey design analysis to deal with correlation within cities. The power of the study was approximately 1 based on the large sample size of the study and different anemia prevalence rates and odds ratios tested. Distribution of different anemia severities and types was also presented as age-specific prevalence rates by sex. Moreover, we created a map to graphically show the prevalence of anemia in study provinces using Microsoft Office Excel. To evaluate the association between anemia and different factors, we used the multivariable Poisson regression analysis with robust variance by applying adjusted prevalence ratio (PR) with 95% confidence interval (CI). Variables of sex, age, job, marital status, cigarettes, hookah, drug use, alcohol use, abdominal obesity, physical activity, diabetes, renal failure, and cancer were selected for adjustment, after identifying their potential confounding effect using the chi-squared test. A p<0.05 was considered statistically significant.

## Results

Out of 163,770 people primarily registered in the cohort study, 2,084 were excluded due to lack of required laboratory data. So, a total of 161,686 participants were finally included for further investigations, of whom 72,387 (44.77%) were male and 89,299 (55.23%) were female.

The mean age was 49.39±9.15 years old. The age group 40–49 years had the greatest number of participants (n = 59,113, 36.56%). Majority of the individuals were married (n = 147,264, 91.08%) and from urban areas (n = 114,424, 70.76%). Table 1 represents the general characteristics of the subjects.

The overall age- and sex-standardized prevalence of anemia was 8.83% (95% CI: 8.70–8.96%) in the included population. The anemia prevalence was also estimated at local scale and represented in Table 2. The highest and the lowest age- and sex-standardized prevalence of anemia pertained to Hormozgan (37.41%, 95% CI: 35.97–38.85%) and Kurdistan (4.57%, 95% CI: 3.87–5.27%) provinces, respectively. Fig 1 represents geographically the age- and sex-standardized prevalence of anemia at regional level.

The crude and age- and sex-standardized, prevalence of anemia have been reported by participants' characteristics in Table 1. The mean hemoglobin concentration was lower in females (13.42±1.36 g/dL) than in males (15.35±1.38 g/dL), and the difference was significant (p<0.001). The age-standardized prevalence of anemia in females was 12.05% (95 CI: 9.73–14.83%), which was notably higher than in males (4.33%, 95 CI: 2.87–6.47%) (Table 1). Also, it was found that the sex-standardized prevalence of anemia was highest in age group 40–49 years (9.39%, 95 CI: 7.62–11.5%) in comparison with other age groups. Fig 2 shows the age-specific prevalence of anemia by sex. As indicated, prevalence of anemia was higher in females than in males in all age groups. In females, we witnessed a slight increasing trend in anemia prevalence from 35–39 years, peaking in 45–49 years, with a moderate decrease in 50–59 years, and an increase thereafter, while there was an increasing trend in males in all age groups.

The age- and sex-standardized prevalence of anemia was higher in rural residents (8.49%), widowed/divorced individuals (10.97%), subjects with primary education (8.91%), housewives (12.26%), those in the third wealth status quartile (8.26%), underweight individuals (11.83%), non-abdominal obese people (9.14%), non-cigarette smokers (9.14%), non-hookah smokers (8.45%), non-drug users (8.40%), non-alcohol users (8.41%), individuals with excessive sleep duration (9.52%), subjects with poor physical activity (8.91%), diabetic individuals (9.02%), hypertensive individuals (8.50%), those with renal failure (17.71%), subjects with cancer (12.75%), individuals with rheumatoid arthritis (9.92%), and those with lupus (12.76%), compared with other subgroups (Table 1).

Multivariable Poisson regression analysis with robust variance showed that being female (PR = 2.97), age group 40–49 years (PR = 1.19), rural residence (PR = 1.24), being retired (PR = 1.53) and housewife (PR = 1.11), third and fourth wealth status quartiles (PR = 1.09 and PR = 1.11, respectively), being underweight (PR = 1.49), drug user (PR = 1.35), inadequate sleep (PR = 1.16), poor physical activity (PR = 1.15), diabetes (PR = 1.09), renal failure (PR = 2.24), and cancer (PR = 1.35) were associated with increased risk of anemia among the cohort population (Table 3). On the other hand, illiteracy (PR = 0.79) and abdominal obesity (PR = 0.77) decreased the risk of anemia.

In Tables 4 and 5, we reported the age-specific prevalence of different anemia types and severities by sex. The most common type of anemia in both sexes and all ages was hypochromic-microcytic anemia (64.46%), followed by normochromic-normocytic (23.12%), and hypochromic-normocytic (7.21%) anemia. Also, the prevalence of mild, moderate and severe anemia was 68.86%, 29.63% and 1.51% in both sexes and all ages.

## Discussion

In the current study, we analyzed the prevalence of anemia among the Iranian adult population using the data derived from PERSIAN cohort study. The overall prevalence of anemia was 8.83%, which is classified as mild public health significance according to the WHO

**Table 1. Prevalence of anemia by characteristics of participants.**

| Variables | Anemia frequency/Study population | Anemia | |
|---|---|---|---|
| | | Crude prevalence (% [95% CI]) | Age- and sex-standardized prevalence (% [95% CI]) |
| **Demographic** | | | |
| **Sex** | | | |
| Male | 3,452/72,387 | 4.77 (4.61–4.92) | 4.33 (2.87–6.47)[†] |
| Female | 10,843/89,299 | 12.14 (11.93–12.36) | 12.05 (9.73–14.83)[†] |
| **Age groups (years)** | | | |
| 35–39 | 2,414/27,192 | 8.88 (8.54–9.22) | 7.92 (5.86–10.62)[‡] |
| 40–49 | 6,104/59,113 | 10.33 (10.08–10.57) | 9.39 (7.62–11.5)[‡] |
| 50–59 | 3,609/48,539 | 7.44 (7.20–7.67) | 6.76 (4.97–9.12)[‡] |
| ≥60 | 2,168/26,842 | 8.08 (7.75–8.40) | 7.68 (5.38–10.85)[‡] |
| **Residence** | | | |
| Urban | 10,307/114,424 | 9.01 (8.84–9.17) | 6.87 (4.73–9.87) |
| Rural | 3,988/47,262 | 8.44 (8.19–8.69) | 8.49 (6.20–11.51) |
| **Marital status** | | | |
| Single | 327/3,368 | 9.71 (8.71–10.71) | 8.90 (6.31–12.43) |
| Married | 12,755/147,264 | 8.66 (8.52–8.80) | 7.93 (6.15–10.17) |
| Widowed and/or divorced | 1,213/11,054 | 10.97 (10.39–11.56) | 10.97 (8.56–13.94) |
| **Socioeconomics variables** | | | |
| **Educational level**[a] | | | |
| Illiterate | 2,933/33,254 | 8.82 (8.52–9.12) | 7.71 (6.13–9.63) |
| Primary | 5,712/58,787 | 9.72 (9.48–9.96) | 8.91 (6.73–11.69) |
| Secondary | 4,236/50,253 | 8.43 (8.19–8.67) | 7.99 (6.26–10.13) |
| Tertiary | 1,409/19,233 | 7.33 (6.96–7.69) | 7.31 (4.49–11.68) |
| **Occupational status**[b] | | | |
| Unemployed | 1,777/21,098 | 8.42 (8.05–8.81) | 8.74 (8.11–9.42) |
| Working | 4,354/72,101 | 6.04 (5.86–6.21) | 5.48 (3.87–7.71) |
| Retired | 598/8,434 | 7.09 (6.54–7.64) | 6.95 (3.96–11.91) |
| Housewife | 7,522/59,286 | 12.69 (12.42–12.96) | 12.26 (9.64–15.47) |
| **Wealth status quartiles**[c] | | | |
| 1st (poorest) | 3,732/40,149 | 9.30 (9.01–9.58) | 8.09 (6.07–10.69) |
| 2nd | 3,503/39,604 | 8.85 (8.57–9.12) | 8.13 (6.44–10.20) |
| 3rd | 3,895/44,169 | 8.82 (8.55–9.08) | 8.26 (6.45–10.52) |
| 4th (richest) | 3,126/37,182 | 8.41 (8.13–8.69) | 8.04 (5.41–11.80) |
| **Individual factors** | | | |
| **BMI categories**[d] | | | |
| Underweight | 378/2,937 | 12.87 (11.66–14.08) | 11.83 (8.27–16.64) |
| Normal | 3,832/41,707 | 9.19 (8.91–9.47) | 8.40 (6.29–11.13) |
| Overweight | 5,374/65,680 | 8.18 (7.97–8.39) | 7.44 (5.74–9.59) |
| Obese | 4,663/50,756 | 9.19 (8.94–9.44) | 8.59 (6.81–10.79) |
| **Abdominal obesity** | | | |
| No | 3,079/33,024 | 9.32 (9.01–9.64) | 9.14 (7.05–11.77) |
| Yes | 11,216/128,662 | 8.72 (8.56–8.87) | 7.79 (6.04–9.98) |
| **Ever cigarette smoker**[e] | | | |
| No | 12,413/126,065 | 9.85 (9.68–10.01) | 9.14 (7.19–11.56) |
| Yes | 1,841/35,024 | 5.26 (5.02–5.49) | 4.76 (3.21–6.99) |
| **Ever hookah smoker**[f] | | | |
| No | 13,367/147,009 | 9.09 (8.95–9.24) | 8.45 (6.62–10.73) |
| Yes | 888/14,065 | 6.31 (5.91–6.72) | 5.39 (3.33–8.60) |

*(Continued)*

**Table 1.** (Continued)

| Variables | Anemia frequency/Study population | Anemia | |
|---|---|---|---|
| | | Crude prevalence (% [95% CI]) | Age- and sex-standardized prevalence (% [95% CI]) |
| **Ever drug user[g]** | | | |
| No | 13,199/144,829 | 9.11 (8.97–9.26) | 8.40 (6.48–10.81) |
| Yes | 1,055/16,259 | 6.49 (6.11–6.87) | 6.15 (4.46–8.44) |
| **Ever alcohol user[h]** | | | |
| No | 13,782/150,796 | 9.14 (8.99–9.28) | 8.41 (6.57–10.71) |
| Yes | 471/10,288 | 4.58 (4.17–4.98) | 4.52 (2.72–7.41) |
| **Sleeping duration[f]** | | | |
| Inadequate | 5,315/56,639 | 9.38 (9.14–9.62) | 8.43 (5.98–11.77) |
| Enough | 7,893/93,841 | 8.41 (8.23–8.59) | 7.81 (6.19–9.82) |
| Excessive | 1,051/10,594 | 9.92 (9.35–10.49) | 9.52 (7.46–12.07) |
| **Physical activity[i]** | | | |
| Good | 4,299/56,521 | 7.61 (7.39–7.82) | 6.94 (5.32–9.02) |
| Poor | 9,954/104,538 | 9.52 (9.34–9.70) | 8.91 (6.81–11.56) |
| **Medical history** | | | |
| **Diabetes[j]** | | | |
| No | 12,070/138,685 | 8.70 (8.55–88.5) | 8.01 (6.17–10.31) |
| Yes | 2,189/22,451 | 9.75 (9.36–10.14) | 9.02 (7.20–11.24) |
| **Hypertension[j]** | | | |
| No | 11,200/127,372 | 8.79 (8.64–8.95) | 8.05 (6.23–10.33) |
| Yes | 3,059/33,764 | 9.06 (8.75–9.37) | 8.50 (6.54–10.98) |
| **Renal failure[j]** | | | |
| No | 14,005/159,802 | 8.76 (8.63–8.90) | 8.05 (6.23–10.34) |
| Yes | 254/1,334 | 19.04 (16.93–21.15) | 17.71 (12.65–24.22) |
| **Cancer** | | | |
| No | 14,113/160,360 | 8.80 (8.66–8.94) | 8.08 (6.27–10.36) |
| Yes | 182/1,326 | 13.73 (11.92–15.69) | 12.75 (9.83–16.34) |
| **Rheumatoid arthritis[j]** | | | |
| No | 13,275/151,613 | 8.75 (8.61–8.89) | 8.03 (6.22–10.32) |
| Yes | 984/9,523 | 10.33 (9.72–10.96) | 9.92 (6.82–14.22) |
| **Lupus[j]** | | | |
| No | 14,239/160,997 | 8.84 (8.71–8.98) | 8.13 (6.31–10.41) |
| Yes | 20/139 | 14.38 (9.01–21.34) | 12.76 (7.71–20.40) |

[†] Standardized for age.

[‡] Standardized for sex.

**Abbreviations:** CI, confidence interval; BMI, body mass index.

**Missing values, count:**

[a], n = 159;

[b], n = 770;

[c], n = 582;

[d], n = 606;

[e], n = 597;

[f], n = 612;

[g], n = 598;

[h], 602;

[i], n = 627;

[j], n = 550.

**Table 2. Total and sex-specific crude and age-standardized prevalence of anemia by location.**

| Location | Crude prevalence (% [95% confidence interval]) | | | Age-standardized prevalence (% [95% confidence interval]) | | |
|---|---|---|---|---|---|---|
| | **Total** | **Male** | **Female** | **Total**[*] | **Male** | **Female** |
| **Iran** | 8.84 (8.70–8.98) | 4.77 (4.61–4.92) | 12.14 (11.93–12.36) | 8.83 (8.70–8.96) | 4.71 (4.56–4.87) | 12.06 (11.85–12.28) |
| **Ardabil** | 5.91 (5.59–6.23) | 1.93 (1.65–2.20) | 9.28 (8.74–9.81) | 5.89 (5.58–6.21) | 1.97 (1.68–2.25) | 9.09 (8.56–9.61) |
| **Chaharmahal and Bakhtiari** | 5.75 (5.29–6.21) | 3.14 (2.64–3.64) | 8.07 (7.33–8.81) | 5.84 (5.37–6.31) | 3.12 (2.62–3.62) | 8.05 (7.31–8.79) |
| **East Azerbaijan** | 4.95 (4.61–5.30) | 1.61 (1.31–1.91) | 7.66 (7.08–8.23) | 4.95 (4.61–5.29) | 1.59 (1.29–1.88) | 7.69 (7.12–8.26) |
| **Fars** | 8.18 (7.85–8.52) | 5.32 (4.92–5.74) | 10.56 (10.06–11.08) | 8.18 (7.88–8.55) | 5.29 (4.89–5.70) | 10.59 (10.08–11.10) |
| **Guilan** | 13.17 (12.52–13.81) | 9.23 (8.42–10.04) | 16.58 (15.61–17.56) | 13.30 (12.65–13.95) | 9.21 (8.41–10.02) | 16.63 (15.66–17.61) |
| **Hormozgan** | 38.14 (36.64–39.64) | 22.66 (20.67–24.64) | 49.57 (47.54–51.60) | 37.41 (35.97–38.85) | 23.01 (20.98–25.03) | 49.14 (47.11–51.17) |
| **Kerman** | 8.92 (8.37–9.49) | 4.43 (3.83–5.02) | 12.86 (11.96–13.76) | 9.25 (8.68–9.82) | 4.35 (3.76–4.93) | 13.3 (12.37–14.22) |
| **Kermanshah** | 9.19 (8.62–9.75) | 5.19 (4.57–5.83) | 12.78 (11.88–13.68) | 9.46 (8.88–10.05) | 5.53 (4.84–6.23) | 12.66 (11.76–13.55) |
| **Khouzestan** | 10.97 (10.35–11.58) | 4.97 (4.30–5.64) | 15.00 (14.10–15.91) | 10.48 (9.90–11.06) | 5.04 (4.36–5.72) | 14.91 (14.01–15.81) |
| **Kohgiluyeh and Boyer-Ahmad** | 6.24 (5.41–7.08) | 3.74 (2.74–4.74) | 8.14 (6.89–9.40) | 6.09 (5.28–6.90) | 3.40 (2.49–4.31) | 8.29 (7.01–9.56) |
| **Kurdistan** | 4.64 (3.94–5.34) | 2.33 (1.57–3.10) | 6.42 (5.33–7.51) | 4.57 (3.87–5.27) | 2.44 (1.63–3.26) | 6.30 (5.22–7.38) |
| **Mazandaran** | 13.37 (12.71–14.03) | 8.59 (7.73–9.44) | 16.62 (15.69–17.56) | 13.00 (12.36–13.64) | 8.31 (7.47–9.14) | 16.83 (15.89–17.77) |
| **Razavi Khorasan** | 9.83 (8.93–10.73) | 6.63 (5.50–7.76) | 12.40 (11.06–13.74) | 9.67 (8.78–10.55) | 6.48 (5.38–7.58) | 12.26 (10.93–13.59) |
| **Sistan and Balouchestan** | 6.79 (6.29–7.28) | 4.42 (3.77–5.06) | 8.31 (7.61–9.00) | 6.48 (6.01–6.96) | 4.15 (3.52–4.78) | 8.38 (7.69–9.08) |
| **West Azerbaijan** | 5.21 (4.60–5.82) | 2.62 (1.96–3.30) | 7.20 (6.25–8.14) | 5.23 (4.62–5.84) | 2.69 (2.01–3.38) | 7.29 (6.33–8.25) |
| **Yazd** | 7.63 (7.11–8.16) | 2.22 (1.81–2.63) | 13.11 (12.16–14.06) | 7.98 (7.44–8.51) | 2.27 (1.85–2.69) | 12.62 (11.7–13.53) |

[*] Standardized for age and sex.

population-based classification [6]. The anemia prevalence estimated in our study is almost in agreement with some previous regional surveys in Iran, in which the anemia prevalence was reported as about 10% [18, 19]. Moreover, anemia was more prevalent in females (12.06%) than in males (4.71%) in our study, which is mainly explained by iron deficiency anemia due to menstruation or pregnancy [20]. The prevalence of anemia increased by age in males, which could be related to chronic diseases and/or iron deficiency anemia [21]. Also, the most common type of anemia in the present population was hypochromic-microcytic anemia, suggesting iron deficiency, chronic diseases, and/or thalassemia as the most likely potential causes of anemia [22, 23].

Among the study provinces, Hormozgan and Kurdistan had the highest and lowest prevalence of anemia, respectively. One of the reasons for the high prevalence of anemia in Hormozgan could be due to the considerable prevalence of beta-thalassemia in this province [24]. Overall, thalassemia with a prevalence of about 10% in several provinces of Iran is one of the most common etiologies of microcytic anemia. However, due to the national thalassemia prevention and treatment program implemented in 1995, Iran has made great progress in prevention and control of this disease [25]. So far, no comprehensive studies have been done to assess the prevalence of anemia in the study provinces, and therefore, our findings opened new windows on this subject for health professionals and authorities to take appropriate measures to identify and manage the cases of anemia in high-prevalence areas.

In this study, we also attempted to evaluate a number of factors potentially linked with anemia. In this regard, comparison of urban and rural data showed that the prevalence rate of anemia was significantly higher individuals living in rural areas. It is expectable that rural population is potentially at a higher risk of anemia due to health disparities and fertility preferences [9]. Moreover, among social habits, we only found a significant association between anemia and drug users, whereas cigarettes, hookah, and alcohol were not identified as anemia

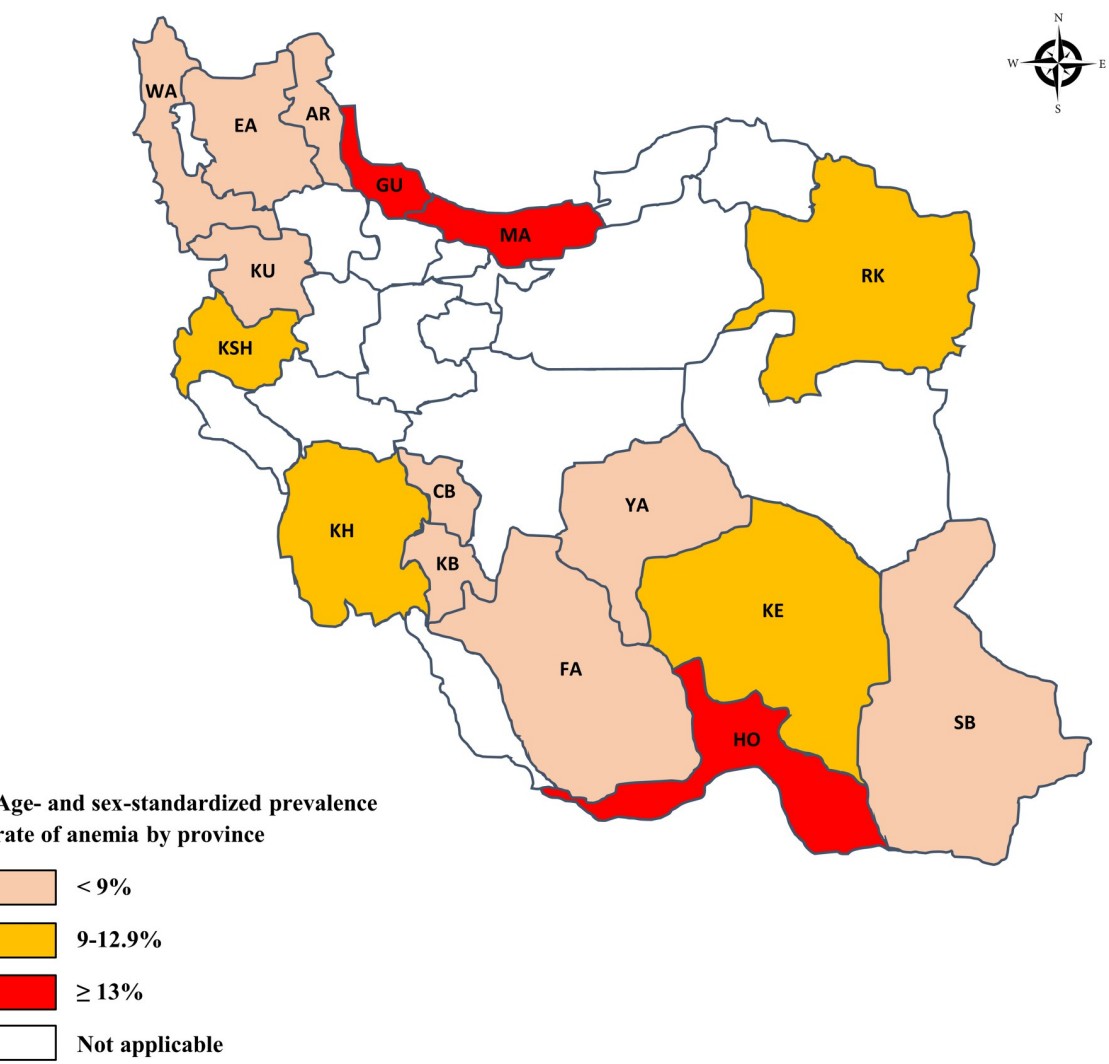

**Fig 1. Graphical presentation of age- and sex-standardized prevalence of anemia in study provinces, including Ardabil (AR), Chaharmahal and Bakhtiari (CB), East Azerbaijan (EA), Fars (FA), Guilan (GU), Hormozgan (HO), Kerman (KE), Kermanshah (KSH), Khouzestan (KH), Kohgiluyeh and Boyer-Ahmad (KB), Kurdistan (KU), Mazandaran (MA), Razavi Khorasan (RK), Sistan and Balouchestan (SB), West Azerbaijan (WA), and Yazd (YA).**

determinants. According to the literature, heroin and cocaine can disrupt iron regulation and suppress erythropoietic activity [26]. It has also been stated that smoking can be possibly accompanied by different types of anemia, such as hemolytic, megaloblastic or aplastic anemias, by a range of various mechanisms [27]. On the other hand, conflicting results exist on the association between alcohol intake and anemia, that is, some studies declared that chronic alcohol abuse can lead to decreased erythrocyte counts and hemoglobin levels by adverse effects on erythropoiesis [28], while some other surveys did not confirm these findings [29].

Based on the analyses, poor physical activity was significantly associated with increased risk of anemia. It has been stated that exercise can enhance hemoglobulin and red blood cell mass through stimulating the erythropoiesis and improving the hematopoietic microenvironment in the bone marrow [30]. We observed that being underweight is directly associated with the risk of anemia. According to the previous studies, there is a discrepancy between anemia and

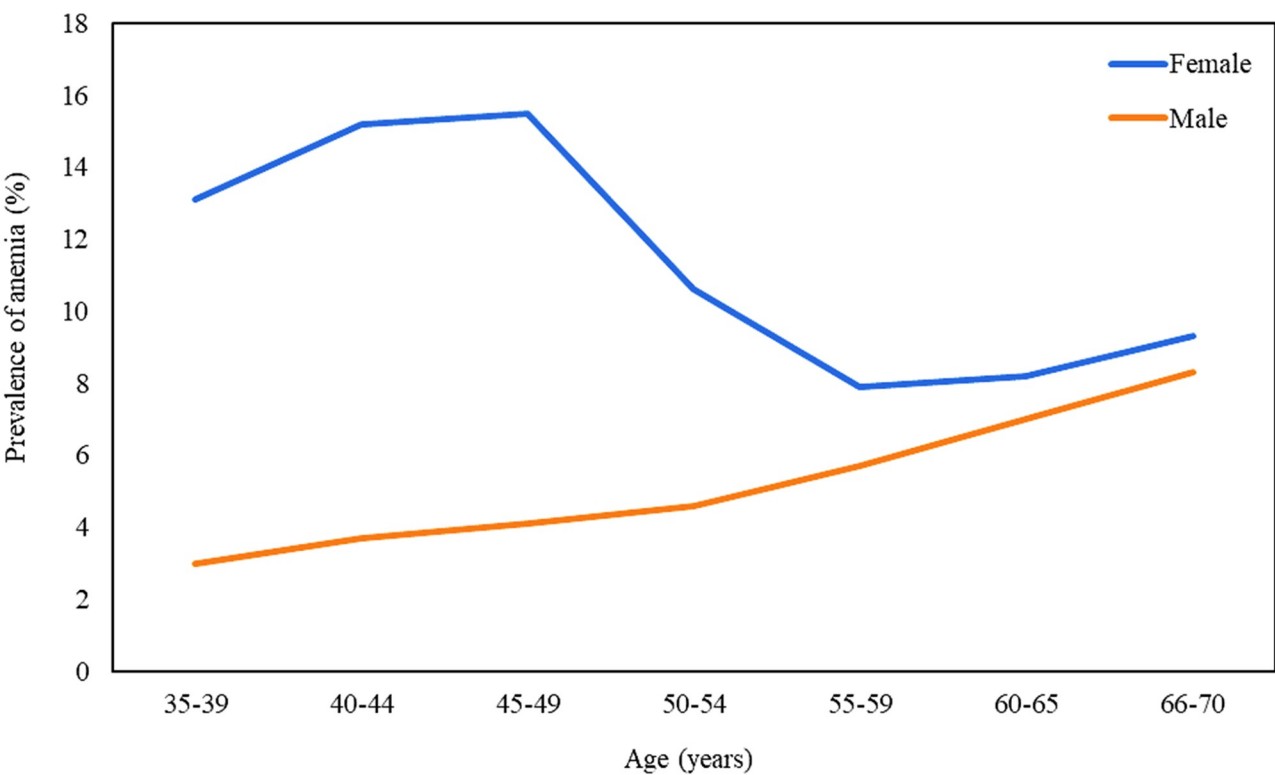

**Fig 2. Age-specific prevalence of anemia by sex.**

weight [31, 32], and more investigations are needed to clarify this association. Our results also demonstrated that inadequate sleep can potentially increase the risk of anemia. There are a few studies on the association between sleep duration and anemia risk, and the findings indicated that either short or long sleep duration might be related with risk of anemia; for example, a recent study reported that both of inadequate and excessive sleep increased the risk of anemia [33].

Regarding past medical history, diabetes, renal failure, and cancer increased the risk of anemia, contrary to hypertension, rheumatoid arthritis, and lupus. In patients with chronic renal failure, the level of erythropoietin (an erythropoietic hormone) is low, leading to decreased red blood cell count [34]. Diabetes can be chronically associated with mild-to-moderate anemia through different ways, such as elevating the level of proinflammatory cytokines like interleukin-6 that has antierythropoietic actions, and reduction of the erythropoietic hormone as a result of nephropathy [35]. Based on the evidence, some autoimmune diseases, such as rheumatoid arthritis and lupus, could also be associated with anemia, unlike the findings of the present study [36, 37].

A strength of this population-based study is the large number of participants included, which could help us to provide representative results with more precise point estimate on the prevalence of anemia. Of course, it should be mentioned that we included adults aged ≥35 years old and our interpretation on study representativeness should be based on this age group. On the other hand, a limitation of the present study was lack of information on some influential factors, such as iron indices, vitamin levels, infections, or genetic disorders, to determine the etiology of anemia. Another limitation is that the present research was based on the respondents' self-report and the collected data might have the likelihood of recall bias.

**Table 3. Univariate and multivariable Poisson regression analysis with robust variance of factors associated with anemia.**

| Variables | Crude prevalence ratio (95% confidence interval) | P-value* | Adjusted prevalence ratio (95% confidence interval) | P-value* |
|---|---|---|---|---|
| **Demographic** | | | | |
| **Sex** | | | | |
| Male | 1 | | 1 | |
| Female | 2.78 (2.64–2.92) | <0.001 | 2.97 (2.74–3.23) | <0.001 |
| **Age groups (years)** | | | | |
| 35–39 | 1 | | 1 | |
| 40–49 | 1.18 (1.11–1.25) | <0.001 | 1.19 (1.12–1.26) | <0.001 |
| 50–59 | 0.85 (0.79–0.91) | <0.001 | 0.83 (0.77–0.89) | <0.001 |
| ≥60 | 0.96 (0.90–1.04) | 0.418 | 0.89 (0.82–0.97) | 0.006 |
| **Residence** | | | | |
| Urban | 1 | | 1 | |
| Rural | 1.23 (1.16–1.30) | <0.001 | 1.24 (1.17–1.32) | <0.001 |
| **Marital status** | | | | |
| Single | 1 | | 1 | |
| Married | 0.89 (0.76–1.03) | 0.139 | 1.14 (0.98–1.33) | 0.089 |
| Widowed and/or divorced | 1.23 (1.04–1.45) | 0.014 | 1.15 (0.97–1.36) | 0.097 |
| **Socioeconomics variables** | | | | |
| **Educational level** | | | | |
| Illiterate | 0.86 (0.81–0.92) | <0.001 | 0.79 (0.74–0.85) | <0.001 |
| Primary | 1 | | 1 | |
| Secondary | 0.89 (0.85–0.92) | <0.001 | 1.02 (0.97–1.08) | 0.339 |
| Tertiary | 0.82 (0.76–0.88) | <0.001 | 1.01 (0.93–1.09) | 0.782 |
| **Occupational status** | | | | |
| Unemployed | 1 | | 1 | |
| Working | 0.62 (0.58–0.67) | <0.001 | 1.08 (0.99–1.17) | 0.067 |
| Retired | 0.79 (0.71–0.88) | <0.001 | 1.53 (1.36–1.72) | <0.001 |
| Housewife | 1.40 (1.31–1.49) | <0.001 | 1.11 (1.04–1.18) | 0.003 |
| **Wealth status quartiles** | | | | |
| 1st (poorest) | 1 | | 1 | |
| 2nd | 1.00 (0.94–1.07) | 0.886 | 1.05 (0.98–1.12) | 0.126 |
| 3rd | 1.02 (0.95–1.08) | 0.506 | 1.09 (1.02–1.16) | 0.007 |
| 4th (richest) | 0.99 (0.93–1.05) | 0.870 | 1.11 (1.03–1.18) | 0.002 |
| **Individual factors** | | | | |
| **BMI categories** | | | | |
| Underweight | 1.40 (1.22–1.62) | <0.001 | 1.49 (1.28–1.73) | <0.001 |
| Normal | 1 | | 1 | |
| Overweight | 0.88 (0.83–0.93) | <0.001 | 0.81 (0.76–0.85) | <0.001 |
| Obese | 1.02 (0.96–1.08) | 0.425 | 0.77 (0.73–0.82) | <0.001 |
| **Abdominal obesity** | | | | |
| No | 1 | | 1 | |
| Yes | 0.85 (0.80–0.89) | <0.001 | 0.77 (0.73–0.81) | <0.001 |
| **Ever cigarette smoker** | | | | |
| No | 1 | | 1 | |
| Yes | 0.52 (0.48–0.55) | <0.001 | 1.00 (0.91–1.09) | 0.958 |
| **Ever hookah smoker** | | | | |
| No | 1 | | 1 | |
| Yes | 0.63 (0.58–0.69) | <0.001 | 0.89 (0.77–1.00) | 0.138 |

(*Continued*)

**Table 3.** (Continued)

| Variables | Crude prevalence ratio (95% confidence interval) | P-value[*] | Adjusted prevalence ratio (95% confidence interval) | P-value[*] |
|---|---|---|---|---|
| **Ever drug user** | | | | |
| No | 1 | | 1 | |
| Yes | 0.73 (0.67–0.79) | <0.001 | 1.35 (1.22–1.48) | <0.001 |
| **Ever alcohol user** | | | | |
| No | 1 | | 1 | |
| Yes | 0.53 (0.47–0.60) | <0.001 | 0.95 (0.84–1.09) | 0.473 |
| **Sleeping duration** | | | | |
| Inadequate | 1.07 (1.03–1.12) | 0.001 | 1.16 (1.11–1.21) | <0.001 |
| Enough | 1 | | 1 | |
| Excessive | 1.21 (1.12–1.32) | <0.001 | 1.04 (0.96–1.13) | 0.278 |
| **Physical activity** | | | | |
| Good | 1 | | 1 | |
| Poor | 1.28 (1.22–1.34) | <0.001 | 1.15 (1.10–1.21) | <0.001 |
| **Medical history** | | | | |
| **Diabetes** | | | | |
| No | 1 | | 1 | |
| Yes | 1.12 (1.06–1.19) | <0.001 | 1.09 (1.03–1.16) | 0.005 |
| **Hypertension** | | | | |
| No | 1 | | 1 | |
| Yes | 1.05 (1.00–1.11) | 0.039 | 0.95 (0.89–1.00) | 0.061 |
| **Renal failure** | | | | |
| No | 1 | | 1 | |
| Yes | 2.20 (1.88–2.57) | <0.001 | 2.24 (1.92–2.62) | <0.001 |
| **Cancer** | | | | |
| No | 1 | | 1 | |
| Yes | 1.57 (1.31–1.89) | <0.001 | 1.35 (1.13–1.63) | 0.001 |
| **Rheumatoid arthritis** | | | | |
| No | 1 | | 1 | |
| Yes | 1.23 (1.14–1.33) | <0.001 | 1.03 (0.94–1.12) | 0.463 |
| **Lupus** | | | | |
| No | 1 | | 1 | |
| Yes | 1.57 (0.88–2.78) | 0.122 | 0.96 (0.51–1.80) | 0.922 |

[*] A p<0.05 was considered statistically significant.

## Conclusion

According to the results, a variable prevalence of anemia was observed across the included provinces. Also, being female, rural residence, being retired and housewife, third and fourth wealth status quartiles, being underweight, drug user, inadequate sleep, poor physical activity, diabetes, renal failure, and cancer, were associated with increased risk of anemia. On the other hand, illiteracy and abdominal obesity decreased the risk of anemia. The present population-based study tried to provide an informative report on the anemia prevalence for health professionals and authorities to take measures for identification and management of the cases of anemia in high-prevalence areas.

**Table 4. Age-specific prevalence of different anemia types by sex (n = 14,295).**

| Sex | Anemia type (n [%]) | | | | | | | |
|---|---|---|---|---|---|---|---|---|
| | Hypochromic-microcytic | P-value[†] | Normochromic-normocytic | P-value[†] | Hypochromic- normocytic | P-value[†] | Others[*] | P-value[†] |
| **Male** | | | | | | | | |
| **All ages** | 2099 (60.81) | <0.001 | 895 (25.93) | <0.001 | 170 (4.92) | 0.517 | 288 (8.34) | <0.001 |
| 35–39 | 265 (76.37) | | 50 (14.41) | | 17 (4.9) | | 15 (4.32) | |
| 40–49 | 698 (69.25) | | 199 (19.74) | | 50 (4.96) | | 61 (6.05) | |
| 50–59 | 686 (60.66) | | 290 (25.64) | | 63 (5.57) | | 92 (8.13) | |
| ≥60 | 450 (46.59) | | 356 (36.85) | | 40 (4.14) | | 120 (12.42) | |
| **Female** | | | | | | | | |
| **All ages** | 7115 (65.62) | <0.001 | 2411 (22.24) | <0.001 | 860 (7.93) | 0.348 | 457 (4.21) | 0.001 |
| 35–39 | 1306 (63.18) | | 501 (24.24) | | 184 (8.9) | | 76 (3.68) | |
| 40–49 | 3566 (69.98) | | 947 (18.58) | | 393 (7.71) | | 190 (3.73) | |
| 50–59 | 1589 (64.12) | | 580 (23.41) | | 191 (7.71) | | 118 (4.76) | |
| ≥60 | 654 (54.41) | | 383 (31.86) | | 92 (7.65) | | 73 (6.08) | |
| **Both sexes** | | | | | | | | |
| **All ages** | 9214 (64.46) | <0.001 | 3306 (23.12) | <0.001 | 1030 (7.21) | 0.033 | 745 (5.21) | <0.001 |
| 35–39 | 1571 (65.08) | | 551 (22.83) | | 201 (8.32) | | 91 (3.77) | |
| 40–49 | 4264 (69.86) | | 1146 (18.77) | | 443 (7.26) | | 251 (4.11) | |
| 50–59 | 2275 (63.04) | | 870 (24.10) | | 254 (7.04) | | 210 (5.82) | |
| ≥60 | 1104 (50.92) | | 739 (34.09) | | 132 (6.09) | | 193 (8.90) | |

[*] Including normochromic-microcytic, normochromic-macrocytic, hyperchromic-microcytic, hyperchromic- normocytic and hyperchromic-macrocytic.

[†] Analyzed by chi-squared test. A p<0.05 was considered statistically significant.

**Table 5. Age-specific prevalence of different anemia severities by sex (n = 14,295).**

| Sex | Anemia severity[*] (n [%]) | | | P-value[†] |
|---|---|---|---|---|
| | Mild | Moderate | Severe | |
| **Male** | | | | 0.162 |
| **All ages** | 3081 (89.25) | 356 (10.31) | 15 (0.44) | |
| 35–39 | 312 (89.91) | 34 (9.80) | 1 (0.29) | |
| 40–49 | 916 (90.87) | 86 (8.53) | 6 (0.60) | |
| 50–59 | 1010 (89.30) | 116 (10.26) | 5 (0.44) | |
| ≥60 | 843 (87.27) | 120 (12.42) | 3 (0.31) | |
| **Female** | | | | <0.001 |
| **All ages** | 6763 (62.37) | 3879 (35.78) | 201 (1.85) | |
| 35–39 | 1310 (63.38) | 731 (35.36) | 26 (1.26) | |
| 40–49 | 3054 (59.93) | 1922 (37.72) | 120 (2.35) | |
| 50–59 | 1566 (63.19) | 862 (34.79) | 50 (2.02) | |
| ≥60 | 833 (69.3) | 364 (30.28) | 5 (0.42) | |
| **Both sex** | | | | <0.001 |
| **All ages** | 9844 (68.86) | 4235 (29.63) | 216 (1.51) | |
| 35–39 | 1622 (67.19) | 765 (31.69) | 27 (1.12) | |
| 40–49 | 3970 (65.04) | 2008 (32.9) | 126 (2.06) | |
| 50–59 | 2576 (71.38) | 978 (27.1) | 55 (1.52) | |
| ≥60 | 1676 (77.31) | 484 (22.32) | 8 (0.37) | |

[*] Mild anemia: hemoglobin 11–12.9 g/dL in males and 11–11.9 g/dL in females; moderate anemia: hemoglobin 8–10.9 g/dL; severe anemia: hemoglobin <8 g/dL.

[†] Analyzed by chi-squared test. A p<0.05 was considered statistically significant.

## Author Contributions

**Conceptualization:** Hossein Poustchi, Farhad Pourfarzi, Vahid Mohammadkarimi, Ayoob Rastegar, Masoumeh Ghoddusi Johari, Mahmood Moosazadeh, Alireza Ostadrahimi, Reza Malekzadeh.

**Data curation:** Farhad Pourfarzi, Fariborz Mansour-Ghanaei, Ali Mousavizadeh, Shideh Rafati, Alizamen Salehifardjouneghani, Iraj Mohebbi, Fatemeh Ezzodini Ardakani.

**Formal analysis:** Mohammad Zamani, Maryam Sharafkhah.

**Investigation:** Mohammad Zamani, Kourosh Noemani, Yahya Pasdar, Anahita Sadeghi.

**Methodology:** Mohammad Zamani, Hossein Poustchi, Amaneh Shayanrad, Mojtaba Farjam, Ebrahim Ghaderi, Fariborz Mansour-Ghanaei, Shideh Rafati, Alizamen Salehifardjoune-ghani, Alireza Khorram.

**Project administration:** Hossein Poustchi, Reza Malekzadeh.

**Resources:** Hossein Poustchi, Reza Malekzadeh.

**Software:** Mohammad Zamani, Maryam Sharafkhah.

**Validation:** Alireza Ostadrahimi, Iraj Mohebbi.

**Visualization:** Amaneh Shayanrad.

**Writing – original draft:** Mohammad Zamani.

**Writing – review & editing:** Hossein Poustchi, Farhad Pourfarzi, Mojtaba Farjam, Kourosh Noemani, Ebrahim Ghaderi, Vahid Mohammadkarimi, Mahmood Kahnooji, Ayoob Raste-gar, Ali Mousavizadeh, Masoumeh Ghoddusi Johari, Mahmood Moosazadeh, Alireza Khorram, Fatemeh Ezzodini Ardakani, Yahya Pasdar, Anahita Sadeghi, Reza Malekzadeh.

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
