## [Decision Letter · Decision Letter 0]

23 Sep 2021

PONE-D-21-27694Prevalence and determinants of anemia in Iran: findings from the PERSIAN cohort studyPLOS ONE

Dear Dr. Malekzadeh

Thank you for submitting your manuscript to PLOS ONE. After careful consideration, we feel that it has merit but does not fully meet PLOS ONE’s publication criteria as it currently stands. Therefore, we invite you to submit a revised version of the manuscript that addresses the points raised during the review process.

This important study was conducted in relatively large population in one part of Iran to assess the prevalence of anemia and related probable risk factors. What was the reason that location selected for this study? As the reviewer mentioned, the study should be design as a cross sectional study with many important preparations such as sample size and power calculation. Since the study focused on age group (>=35 year) please make sure this is pronounced in the manuscript and also why they are important age group.==============================

We look forward to receiving your revised manuscript.

Kind regards,

Hassan Ashktorab

Academic Editor

PLOS ONE

Journal Requirements:

Reviewers' comments:

Reviewer's Responses to Questions

**Comments to the Author**

1. Is the manuscript technically sound, and do the data support the conclusions?

Reviewer #1: Partly

Reviewer #2: No

2. Has the statistical analysis been performed appropriately and rigorously? 

Reviewer #1: No

Reviewer #2: No

3. Have the authors made all data underlying the findings in their manuscript fully available?

Reviewer #1: No

Reviewer #2: No

4. Is the manuscript presented in an intelligible fashion and written in standard English?

Reviewer #1: Yes

Reviewer #2: Yes

5. Review Comments to the Author

Reviewer #1: This study was conducted in relatively large population to assess the prevalence of anemia and related probable risk facors in Iran.

My questions and recommendations are:

This manuscript is a result of a cross-sectional study. please change the title according to STROBE checklist. please specify the study age group (>=35 year) in title.

Regarding sample size, power analysis in pure descriptive cross-sectional prevalence study is not mandatory. but in analytic C.S. study you should calculate the power of the study for comparisons.

Please explain about the method of sex and age standardization.

Please stated the version of STATA software.

Regarding the selection of confounding factors, according to which level of p- value you were selected them? what is the type of selection method? Step-wise method and p-value base criteria are not recommended.

In table 1, because of your study were restricted to a special age group (35 years and older), please correct the age group strata. please calculate and show the amount of p-value for each variables. please clarify which level of category is different significantly? Also specify statistical test in each comparison. In the subtitle, which number in the table does refer to *p value?

How did you manage the missing data?

In analyses of data from cross-sectional studies, the Poisson models with robust variance and calculating prevalence ratio(PR) are better alternatives than logistic regression and OR. Please use this analysis and compare the results with logistic regression analysis.

In table 4, specify the statistical tests in table subtitle. and stated which level of subgroup is different significantly.

Discussion, please clarify all potential study bias in this study and your efforts to decrease them. for example, simultaneous measurement of exposure and outcome may lead to a very important bias and change the behavior of people after awareness of their disease. Were potential confounders identified? and were they managed appropriately in the study design and/or analysis? Please address the major potential cofounders according to literature. Please identify the effect measures are over- exaggerated or not? why? Discuss both direction and magnitude of any potential bias. Describe any efforts to address potential sources of bias. The most important factor which is against the representative results of the your study is evaluation of anemia in a specific age group. How you can solve this limitation?

Reviewer #2: This is a good piece of research, but suffers from some mistakes in the statistical analysis that resulted in wrong interpretation of the result. In particular for logistic regression: 1) it is not clear whether all potential risk factors were included in the analysis; 2) choice of reference group is wrong for some important factors like education and BMI. The reference group for education should be primary school and not "Illiterate''; similarly the reference category for BMI should be ''Normal'' and not underweight. This problem might happened for other risk factors that should be corrected. If you are careful in interpreting the result the current analysis could be OK. However, the conclusion stated that higher educational level increase the risk and higher BMI decrease the risk, these interpretations are inaccurate and misleading due to selection wrong reference category. In fact only underweight individuals are more at risk of anemia based on this data, which is logical. Regarding the education level if you select primary as your reference category then only illiterate will be significant, which you need to find interpretation for that by exploring other risk factors. You need to redo the analysis, that is why I suggested major revision otherwise it is rather few simple corrections in the statistical analysis followed by proper interpretation.

6. PLOS authors have the option to publish the peer review history of their article (what does this mean?). If published, this will include your full peer review and any attached files.

Reviewer #1: No

Reviewer #2: No

---

## [Author Response · Author response to Decision Letter 0]

31 Oct 2021

Authors’ Response to Reviewers’ Comments

Journal title: PLOS ONE

Manuscript title: Prevalence and determinants of anemia among Iranian population aged ≥35 years: A PERSIAN cohort–based cross-sectional study

Manuscript Number: PONE-D-21-27694R1

Dear Dr Chenette and Prof Ashktorab,

Thank you very much for giving us the opportunity to submit a revised version of above manuscript to PLOS ONE journal. We would like to thank the editors and the reviewers for the time taken in reviewing and helping us to improve the paper. The reviewers’ comments have all been addressed. We attach a highlighted revised version of our paper with all alterations highlighted in yellow, a clean version, and a point-by-point response to the reviewers’ comments. All authors agree with its publication and confirm that it is not being considered for publication elsewhere. 

Yours sincerely

On behalf of the authors

Journal Requirements:

Response: The authors appreciate the useful comments. The relevant corrections were made in the newer version of manuscript (font sizes, bibliography, etc.).

Response: Thank you very much for your comment. The study protocol and individual participant data that underlie the results reported in this study, after de-identification (text, tables, and figures) can be shared with investigators whose proposed use of the data has been approved by the independent review committee of Tehran University of Medical Sciences and Digestive Diseases Research Institute. Data can be provided for projects related to the topic of the present study. The proposals should be directed to the PERSIAN cohort center (email: info@persiancohort.com), and/or Digestive Diseases Research Institute (email: info@ddri.ir), and/or Prof Reza Malekzadeh (email: dr.reza.malekzadeh@gmail.com), the senior author of the manuscript and the project leader.

Response: Thank you for your comment. The ORCID for the corresponding author has been verified.

Response: We appreciate your comment. We designed the figures using Microsoft Office Excel and without plagiarism issue as mentioned in the methods, and we certify that no portion of this manuscript (text, figures) has been previously published.

5. Review Comments to the Author

Reviewer #1: This study was conducted in relatively large population to assess the prevalence of anemia and related probable risk facors in Iran.

My questions and recommendations are:

This manuscript is a result of a cross-sectional study. please change the title according to STROBE checklist. please specify the study age group (>=35 year) in title.

Response: Thank you very much for your comment. The title has been revised as per your recommendation.

Regarding sample size, power analysis in pure descriptive cross-sectional prevalence study is not mandatory. but in analytic C.S. study you should calculate the power of the study for comparisons.

Response: Thank you for the point. As per your comment, we have stated that the power of the study was approximately 1 based on the large sample size of the study and different anemia prevalence rates and odds ratios tested (Page 8).

Please explain about the method of sex and age standardization.

Response: We thank the reviewer’s comment. We have stated that the age- and sex-standardized prevalence rates were calculated using direct method, and Iran national census in 2016 were considered as a standard population (Page 8).

Please stated the version of STATA software.

Response: The version has been added.

Regarding the selection of confounding factors, according to which level of p- value you were selected them? what is the type of selection method? Step-wise method and p-value base criteria are not recommended.

Response: We appreciate your comment. We tried to select the potential determinants of anemia on the basis of the literature and availability in our database. Also, we only aimed to assess the association of the covariates with anemia, but not their predictive roles; therefore, we did not use step-wise method. We have mentioned new references in the revised manuscript upon which we chose the factors (Refs 9-11).

In table 1, because of your study were restricted to a special age group (35 years and older), please correct the age group strata. please calculate and show the amount of p-value for each variables. please clarify which level of category is different significantly? Also specify statistical test in each comparison. In the subtitle, which number in the table does refer to *p value?

Response: Thank you for your comment. The age group name has been corrected in all tables. We also added p-values for both of univariate and multivariable analyses in Table 3 as per your comment. A p-value less than 0.05 was considered significant.

How did you manage the missing data?

Response: The missing data were not included in the analyses. Considering that the rate of missing data was relatively very low, their effect would be ignorable.

In analyses of data from cross-sectional studies, the Poisson models with robust variance and calculating prevalence ratio(PR) are better alternatives than logistic regression and OR. Please use this analysis and compare the results with logistic regression analysis.

Response: Thank you for your comment. According to your recommendation, we have redone all of the relevant analyses and revised the main text and Table 3.

In table 4, specify the statistical tests in table subtitle. and stated which level of subgroup is different significantly.

Response: The statistical test (chi-squared test) and the level of significance have been added. We performed new analyses for each subgroup and added p-values in Table 4.

Discussion, please clarify all potential study bias in this study and your efforts to decrease them. for example, simultaneous measurement of exposure and outcome may lead to a very important bias and change the behavior of people after awareness of their disease. Were potential confounders identified? and were they managed appropriately in the study design and/or analysis? Please address the major potential cofounders according to literature. Please identify the effect measures are over- exaggerated or not? why? Discuss both direction and magnitude of any potential bias. Describe any efforts to address potential sources of bias.

Response: We thank you for your comments. About the simultaneous measurement of exposure and outcome, it should be noted that we did not enroll individuals to specifically assess their anemia only. In fact, we collected a list of data from the people for the PERSIAN cohort and one of the categories was hematologic data. Therefore, in the cohort, the individuals were informed not only about the protocol of hematologic data, but also about other variables at the same time. Thus, people were not aware of their disease during data collection. Regarding confounders, as replied earlier, we tried to select the high potential risk factors according to the literature and availability in our database; however, we agree with your concern that one of our limitations was lack of information on some influential factors, such as iron indices, vitamin levels, infections, or genetic disorders, to determine the etiology of anemia, which was mentioned in the limitations. Finally, we have mentioned that the present research was based on the respondents’ self-report and the collected data might have the likelihood of recall bias (Page 21).

The most important factor which is against the representative results of the your study is evaluation of anemia in a specific age group. How you can solve this limitation?

Response: According to your comment, we added this statement in the end of Discussion that we included adult subjects aged ≥35 years old and our interpretation on study representativeness should be based on this age group.

Reviewer #2: This is a good piece of research, but suffers from some mistakes in the statistical analysis that resulted in wrong interpretation of the result. In particular for logistic regression: 1) it is not clear whether all potential risk factors were included in the analysis

Response: Thank you very much for your comment. We tried to select the high potential risk factors according to the literature and availability in our database. In this regard, we chose 22 factors potentially associated with the risk of anemia; however, we agree with your concern that one of our limitations was lack of information on some influential factors, such as iron indices, vitamin levels, infections, or genetic disorders, to determine the etiology of anemia, which was mentioned in the Discussion as a limitation.

2) choice of reference group is wrong for some important factors like education and BMI. The reference group for education should be primary school and not "Illiterate''; similarly the reference category for BMI should be ''Normal'' and not underweight. This problem might happened for other risk factors that should be corrected. If you are careful in interpreting the result the current analysis could be OK. However, the conclusion stated that higher educational level increase the risk and higher BMI decrease the risk, these interpretations are inaccurate and misleading due to selection wrong reference category. In fact only underweight individuals are more at risk of anemia based on this data, which is logical. Regarding the education level if you select primary as your reference category then only illiterate will be significant, which you need to find interpretation for that by exploring other risk factors. You need to redo the analysis, that is why I suggested major revision otherwise it is rather few simple corrections in the statistical analysis followed by proper interpretation.

Response: We appreciate your helpful comment. We changed the reference category for “Residence”, “Educational level”, “BMI categories” and “Sleeping duration”, and re-analyzed the data (Table 3) and corrected the interpretations in the main text.

---

## [Decision Letter · Decision Letter 1]

27 Jan 2022

Prevalence and determinants of anemia among Iranian population aged ≥35 years: A PERSIAN cohort–based cross-sectional study

PONE-D-21-27694R1

Dear Dr. Malekzadeh

We’re pleased to inform you that your manuscript has been judged scientifically suitable for publication and will be formally accepted for publication once it meets all outstanding technical requirements.

Kind regards,

Hassan Ashktorab

Academic Editor

PLOS ONE

Additional Editor Comments (optional):

Reviewers' comments:

Reviewer's Responses to Questions

**Comments to the Author**

1. If the authors have adequately addressed your comments raised in a previous round of review and you feel that this manuscript is now acceptable for publication, you may indicate that here to bypass the “Comments to the Author” section, enter your conflict of interest statement in the “Confidential to Editor” section, and submit your "Accept" recommendation.

Reviewer #1: All comments have been addressed

Reviewer #2: All comments have been addressed

2. Is the manuscript technically sound, and do the data support the conclusions?

Reviewer #1: Yes

Reviewer #2: Yes

3. Has the statistical analysis been performed appropriately and rigorously? 

Reviewer #1: Yes

Reviewer #2: Yes

4. Have the authors made all data underlying the findings in their manuscript fully available?

Reviewer #1: No

Reviewer #2: No

5. Is the manuscript presented in an intelligible fashion and written in standard English?

Reviewer #1: Yes

Reviewer #2: Yes

6. Review Comments to the Author

Reviewer #1: Dear EIC

Thank you very much for your kind invitation to review of this manuscript revision.

I would like to inform you that the authors have reviewed all my comments and responded in a complete and appropriate manner. They have also made the necessary corrections in the text of the manuscript. The manuscript is eligible for publication now.

Sincerely yours

Reviewer #2: There are still few minor corrections. in the Abstract Iran is typed as IrAN.

In statistics we usually talk about power of a test rather than power of a study.

In table 3, you need to mention what Adjusted prevalence

ratio is adjusted for.

7. PLOS authors have the option to publish the peer review history of their article (what does this mean?). If published, this will include your full peer review and any attached files.

Reviewer #1: **Yes: **Ali Reza Safarpour, MD, MPH, Ph D.

Reviewer #2: No

---

## [Editor Report · Acceptance letter]

31 Jan 2022

PONE-D-21-27694R1 

Prevalence and determinants of anemia among Iranian population aged ≥35 years: A PERSIAN cohort–based cross-sectional study 

Dear Dr. Malekzadeh:

I'm pleased to inform you that your manuscript has been deemed suitable for publication in PLOS ONE. Congratulations! Your manuscript is now with our production department. 

Kind regards, 

on behalf of

Dr. Hassan Ashktorab 

Academic Editor

PLOS ONE